# Polymeric Nanoparticles in Gene Therapy: New Avenues of Design and Optimization for Delivery Applications

**DOI:** 10.3390/polym11040745

**Published:** 2019-04-25

**Authors:** Raj Rai, Saniya Alwani, Ildiko Badea

**Affiliations:** Drug Design and Discovery Research Group, College of Pharmacy and Nutrition, University of Saskatchewan, Saskatoon, SK S7N 5E5, Canada; rkr201@mail.usask.ca (R.R.); ssa930@mail.usask.ca (S.A.)

**Keywords:** top down and bottom up synthesis, green chemistry, colloidal stability of polymeric nanoparticles, blood circulation of polymeric nanoparticles, cytotoxicity, cellular internalization of the polymeric nanoparticles, biodistribution

## Abstract

The field of polymeric nanoparticles is quickly expanding and playing a pivotal role in a wide spectrum of areas ranging from electronics, photonics, conducting materials, and sensors to medicine, pollution control, and environmental technology. Among the applications of polymers in medicine, gene therapy has emerged as one of the most advanced, with the capability to tackle disorders from the modern era. However, there are several barriers associated with the delivery of genes in the living system that need to be mitigated by polymer engineering. One of the most crucial challenges is the effectiveness of the delivery vehicle or vector. In last few decades, non-viral delivery systems have gained attention because of their low toxicity, potential for targeted delivery, long-term stability, lack of immunogenicity, and relatively low production cost. In 1987, Felgner et al. used the cationic lipid based non-viral gene delivery system for the very first time. This breakthrough opened the opportunity for other non-viral vectors, such as polymers. Cationic polymers have emerged as promising candidates for non-viral gene delivery systems because of their facile synthesis and flexible properties. These polymers can be conjugated with genetic material via electrostatic attraction at physiological pH, thereby facilitating gene delivery. Many factors influence the gene transfection efficiency of cationic polymers, including their structure, molecular weight, and surface charge. Outstanding representatives of polymers that have emerged over the last decade to be used in gene therapy are synthetic polymers such as poly(l-lysine), poly(l-ornithine), linear and branched polyethyleneimine, diethylaminoethyl-dextran, poly(amidoamine) dendrimers, and poly(dimethylaminoethyl methacrylate). Natural polymers, such as chitosan, dextran, gelatin, pullulan, and synthetic analogs, with sophisticated features like guanidinylated bio-reducible polymers were also explored. This review outlines the introduction of polymers in medicine, discusses the methods of polymer synthesis, addressing top down and bottom up techniques. Evaluation of functionalization strategies for therapeutic and formulation stability are also highlighted. The overview of the properties, challenges, and functionalization approaches and, finally, the applications of the polymeric delivery systems in gene therapy marks this review as a unique one-stop summary of developments in this field.

## 1. Introduction

The field of polymeric nanoparticles (PNP) is quickly expanding and playing a pivotal role in a wide spectrum of areas from electronics, photonics, conducting materials, and sensors to pollution control, environmental technology, and medicine [1,2,3]. Gene therapy is a relatively new area of medicine that is able to alleviate and cure many diseases that are unable to be mitigated by traditional medicine, and PNPs could play a crucial role in advancing this field. Recently, US Food and Drug Administration (FDA) approved Alnylam’s siRNA drug Onpattro for hereditary amyloidosis. Onpattro encapsulates the therapeutic siRNA moiety into a lipid nanoparticle (NP), delivering the drug directly to the liver via an infusion and preventing the body from producing disease-causing proteins [4]. However, significant efforts are still needed to overcome barriers for gene delivery. Viral vectors, while efficient, pose safety issues, despite the continuous efforts of virologists to minimize their immunogenicity and side effects. Non-viral delivery systems entered the scene because of their low toxicity, potential for targeted delivery, long-term stability, lack of immunogenicity, and relatively low production cost [1].

Cationic lipid-based non-viral gene delivery systems were the first type of non-viral systems [5]. They interact with the negatively charged phosphate groups present in nucleic acids via electrostatic forces to form nanoparticles (NPs), called lipoplexes. Lipoplexes are able to protect their genetic cargo from degradation, and deliver inside mammalian cells. This breakthrough opened the opportunity for other non-viral vectors, such as polymers. Cationic polymers are promising candidates for non-viral gene delivery due to their facile synthesis and flexible properties [6]. Polymers can be conjugated with genetic material, forming polyplexes at physiological pH, to facilitate gene delivery [2,7]. There are many factors that influence the gene transfection efficiency of cationic polymers, i.e., structure, molecular weight, and surface charge. Outstanding representatives of polymers to be used in gene therapy are synthetic polymers such as poly(l-lysine), poly(l-ornithine), linear and branched polyethyleneimine, diethylaminoethyl-dextran, poly(amidoamine) dendrimers, and poly(dimethylaminoethyl methacrylate [3,8,9,10,11,12]. Natural polymers, such as chitosan, dextran, and gelatin, and complex synthetic designs like guanidinylated bioreducible polymer have also been explored.

This review presents a comprehensive assessment of polymers used in gene therapy (Figure 1), connecting a range of topics from their synthesis to practical applications. Summaries provided here regarding synthesis, formulation design, functionalization, and therapeutic applications for non-invasive gene delivery utilizing PNPs makes this review a unique one-stop summary for many developments emerging in this field.

## 2. Properties of Polymeric Nanoparticles Advantageous for Biomedical Use

Polymers are some of the most common materials studied as nanocarriers for drug and gene delivery. This is due to numerous innate properties, such as the versatility of structural conformations, biodegradability, and ease of synthesis, which have served to be beneficial for the design of PNPs.

### 2.1. Biodegradability

Mankind has been provided with many naturally occurring polymers, which we have utilized over time in our daily living. Nature has revealed many building blocks laying the foundation for the synthetic generation of commonly used polymers in nanoscience. Common organic compounds like cellulose or lignin being the most abundant biopolymers on earth has defined the possibility of versatile structural and compositional conformations. Due to this diversity, it is easy to modify the structures of PNPs to package and deliver cargo, genetic material in this review, to the desired site. There have been many studies over the years that describes tailored modification of polymers to allow gene binding and protection. Amine-functionalized, diene-based polymers poly [2-(*N*,*N*-diethylaminomethyl)-1,3-butadiene] exhibited low cytotoxicity and high transfection efficiency to deliver plasmid deoxyribonucleic acid (pDNA) [13]. Polymers have also been functionalized with tumor-targeting peptides to direct the delivery of genetic materials specifically to the malignant cells. One such designed technology is the linear-dendritic hybrid polymer with a peptide that targets glucose-regulated antigens (protein-78 kDa) on human cancer cells [14].

Most of the new technologies in polymeric gene delivery targets two major aspects simultaneously, which are: (1) optimum gene delivery in the targeted cells and (2) minimal retention of the delivery vehicle in the body to quench toxic effects. Natural or synthetic biodegradable polymers with various functionalization designs perform this function well. The major advantage of biodegradable NPs versus non-biodegradable counterparts is the prevention of toxicity and the ease of elimination, i.e., no concern regarding the accumulation after repeated administration [15]. This property is unique to PNPs and many studies have utilized this property to target complex disease sites including the brain [16]. The science of biodegradable NPs encompasses the use of proteins, polysaccharides, and synthetic polymers as well. Synthetic polymers like poly-lactic-acid (PLA), poly-d-l-glycolide (PLG), poly-d-l-lactide-co-glycolide (PLGA), and poly-cyanoacrylate (PCA) are commonly used to generate biodegradable PNPs [16]. For example, PLGA degrades via the hydrolysis of its ester linkages in the presence of water, generating lactic acid and glycolic acid, both being natural metabolites in the body [17]. The rate of degradation can be controlled by optimizing the number of lactic and glycolic acid monomers. Due to the presence of a methyl group, the increase in lactic acid content makes PLGA more hydrophobic and slows down its degradation in the water-rich body environment [18]. Moreover, molecular weight, polydispersity index (PDI), and type of the encapsulated cargo also affects the biodegradability of PNPs [16,19]. Biodegradability of PLGA NPs has been put into practice as drug delivery implants for brain tumors, and have been successful in reaching the clinical trials stage of studies [20].

Another example of biodegradable polymer used in nanomedicine is PLA, in which the carboxylic acid hydrolyzes and contributes to the acid-catalyzed hydrolysis [21]. It hydrolyses in lactic acid, which in turn converts into glucose and is used as a source of energy for metabolic functions. Since PLA has a minimal risk of accumulation at the target site, it is a good candidate for drug delivery to targets like hair follicles and sebaceous glands [22]. Another interesting candidate in this area is PCA, the polymer that is intrinsically unstable and can be easily degraded by coming in contact with water, elevated temperature, or even in basic solution [23]. The degradation mechanism involves base-catalyzed unzipping from the chain terminus and then following the re-equilibration of the chains. The polymer has a slower rate of degradation as compared to polylactides and are used for generating long-term implantable devices [16].

### 2.2. Facile Chemistry

#### 2.2.1. Versatality of Functionalization

Common polymers like PLGA are soluble in a wide range of common solvents including dichloromethane, acetone, ethyl acetate, or tetrahydrofuran, which makes the functionalization and other chemical modifications fast and easy [24]. The stability of polyethylene glycol (PEG), another polymer used in nanomedicine under variable synthetic conditions, has been employed to perform facile functionalization of soft and hard NPs. PEG remains stable at high temperature and the sonication performed during preparation of the particles [25]. Solubility of PEG in different solvents has opened new avenues to target this approach for many types of NPs.

#### 2.2.2. Ease of Synthesis

PNPs of different sizes can be generated for specific applications under mild conditions. One such example is the mini-emulsion polymerization technique where PNPs were generated at room temperature with minimal chemical and photo-stress to carry out encapsulation of sensitive imaging dyes [26]. These techniques are detailed in the next chapter. In addition, research is expanding regarding creating environmentally friendly PNPs as part of the “green nanomedicine” movement, a recent approach to make use of technologies that are environmentally friendly and has its base in nature surrounding us [27,28]. To expand biological drug delivery vehicles based on proteins and lipids, the development of NPs from natural origins or through utilizing environmentally friendly chemical reactions has opened a new chapter in drug/gene delivery science regarded as “green nanotechnology.” In terms of polymer science, this area mainly encompass two major focuses: (1) the production of NPs either using natural polymers or green chemistry and (2) the coating of other organic/inorganic NPs with natural polymers to improve their innate biocompatibility [28]. An example of green nanomedicine using PNPs is the creation of a nanocomposite consisting of polyethylenimine-grafted chitosan oligosaccharide with hyaluronic acid and small interfering RNA (siRNA) to target gene therapy for endometriosis. This formulation effectively delivered the therapeutic gene in CD44 cells, and significantly reduced the clinical signs of endometriosis in tissues. The system was also biocompatible towards reproductive organs [29].

Another aspect of green chemistry applied to polymers is their utilization for coating organic and inorganic NPs to enhance the biocompatibility and targeting, as well as for reducing trafficking and degradation during circulation. Among such applications, the development of pH sensitive hollow mesoporous silica NPs is noteworthy. The NPs were coated first with cyclodextrins (CDs) followed by conjugation of PEG via a linking moiety called adamantine. This design introduced new insights into green chemistry-derived synthesis using simple pH-responsive chemical linkages. The bonds between different entities in the NPs were built such that they utilize natural cleavage mechanisms at pH changes in the tumor microenvironment. Moreover, the chemicals and solvents used in fabrication correspond to the criteria for green-chemistry [30].

Table 1 summarizes some major aspects of green nanotechnology [31] and their practical implication with respect to polymer science.

### 2.3. Scalable Production

Due to extensive research performed on PNPs over time and well-established proof-of-concepts, many new procedures arose for their large-scale industrial production. These innovations focus on reducing the inter-batch variations in the particle properties and making the process cost- and labor-effective. Two ways are presented in the literature for improving the large-scale production of PNPs: (1) process automation via developing sophisticated equipment and (2) tailoring synthetic procedures for easy reproducibility on an industrial scale.

The first approach includes automation units like the Nanoassemblr^®^ platform designed by precision nanosystems [33]. Automation has been applied in pre-existing technology for producing PNPs [34]. This approach has been applied for designing non-spherical anisotropic PNPs as a delivery vehicle for genetic materials [35,36,37,38]. In many cases, anisotropic particles are designed via an expensive top-down approach called particle replication in a non-wetting template or PRINT, a technique that copies the shape as presented on the photolithographic mask [39]. Another technique producing non-spherical PNPs, which is relatively cheaper, is a thin-film stretching method where polymers are adhered to the glass surface. The conventional form of this technique utilizes a manual control that is time-consuming and is unable to maintain uniform strain across the field leading to many inconsistencies. Hence, its industrial application is very limited. This study improved the process by introducing an automated thin-film stretcher, resulting in the ability to control particle size distribution. Moreover, a controlled strain rate in two dimensions yielded ellipsoid particles of various diameters. The technique proved to be successful in designing many types of PNPs including PLGA, PCL, and PLGA/PBAE (poly-beta amino esters) hybrids [34].

Another strategy to improve scalability encompass modifications of chemical methods to produce PNPs without compromising the core particle properties. One such technique providing scalable production of conjugated PNPs for bioimaging is nanoprecipitation or mini-emulsion method. Nanoprecipitation uses very dilute polymer solutions which results in solid content typically less than 500 ppm, whereas mini-emulsion is a multi-step procedure, making it time-consuming and difficult to maintain consistency. They utilized Suzuki aryl-aryl coupling of conjugated polymers with the aid of an ionic surfactant to maintain emulsion stability. The resulting conjugated PNPs were fluorescent and easily detected in flow cytometry after just two hours of incubation. Moreover, they were not cytotoxic for an extended incubation time of 24 h. Since the technology is simple and the resulting particles are pure, it is more scalable than previously employed methods [40]. Earlier, another group suggested a method to produce biodegradable PNPs consisting of copolymers of PLGA and PEG via a scalable emulsification method using low molecular weight emulsifiers. Simply, the replacement of emulsifier from high molecular weight to low molecular weight enabled these NPs to possess mucus-penetrating properties rather than mucus-adhesive properties [41]. They were found to penetrate in undiluted human mucus, opening avenues for designing treatments to target respiratory airways, gastrointestinal tract, and female reproductive organs. This technology particularly focused on preparing NPs to encapsulate proteins, nucleic acids, and peptides that are difficult to formulate via a nanoprecipitation method [41]. On the other hand, the emulsification method is straightforward and industrially applicable.

## 3. Strategies for Designing Polymeric Nanoparticles

The overall efficiency of a gene delivery vehicle depends on many factors like entrapment efficiency, particle size, and surface chemistry of the NP/gene complexes. The process parameters involved in synthesis also play a major role. Each synthetic approach is unique in providing specific entrapment efficiency for water-soluble or water-insoluble therapeutic moieties. Moreover, most PNPs are designed in house for applications including gene therapy. Therefore, it is imperative to discuss these synthetic approaches utilized for designing common polymeric gene carriers.

The following section summarizes some traditional and novel approaches providing reference to their suitability for water-soluble therapeutic moieties mainly genetic materials. PNPs are prepared either using a top-down strategy (involves milling of pre-made polymers to appropriately sized particles) or using a bottom up strategy (requires direct polymerization of monomers using conventional poly-reactions), summarized in Figure 2 [42]. The review briefly confers the routes to duplicate polymers of natural origin such as chitin/chitosan, dextran, and their derivatives. Chitin is composed of β (1→4)-linked 2-acetamido-2-deoxy-β-D-glucose (*N*-acetylglucosamine) making it soluble for a few dilute organic acids and inorganic acids [43]. Chitosan and chitosan derivatives are most commonly synthesized via deacetylation of chitin and further derivatization. Zhang et al. synthesized two water-soluble chitin and chitosan derivatives, namely O-(2-hydroxy-3-trimethylammonium) propyl chitin (OHT-chitin) and *N*-(2-hydroxy-3-trimethylammonium) propyl chitosan (NHT-chitosan). Similarly, another derivative, polyethylenimine-graft-chitosan (PEI-g-chitosan), was synthesized via performing the cationic polymerization of aziridine in the presence of water-soluble oligo-chitosan as a novel gene delivery system [44]. Dextran, another polymer, was used to synthesize dextran-spermine cationic polysaccharide via the reductive amination between oxidized dextran and natural oligoamine spermine [45]. Recently published reviews elaborate various aspects of synthesis and the design of natural polymers in gene therapy [46,47].

### 3.1. Top-Down Strategy for Polymer Synthesis

Top down strategies utilize pre-formed polymers to generate PNPs using solvent evaporation, salting-out dialysis, nanoprecipitation, and supercritical fluid technology, singly or in combination.

#### 3.1.1. Solvent Evaporation Method

Geckeler et al. developed the method of PNP preparation from pre-formed polymers using solvent evaporation (Figure 3) [42]. The method calls for emulsion formulation using polymer solutions in volatile solvents.

The NP dispersion is generated from the emulsion by evaporating the solvent from polymer and diffusing it through the continuous phase [48,49]. Conventionally, there are two approaches to form emulsions: (1) the preparation of single-emulsions, e.g., oil-in-water, or (2) more complex double-emulsions such as (water-in-oil)-in-water [50]. Zambaux et al. synthesized poly(lactic acid) (PLA) PNPs with an average size of 200 nm and a low PDI (<0.1) using dichloromethane and polyvinyl alcohol (PVA) as the solvent and stabilizing agent, respectively [51]. The particle size in this method is dependent on the type and concentration of stabilizer, homogenizer speed, and polymer concentration used in the process [52].

#### 3.1.2. Solvent Displacement Method

This method, also known as nanoprecipitation, consists of precipitating the pre-formed polymer from an organic solution and then diffusing the organic solvent in an aqueous medium with or without a surfactant (Figure 4) [54,55,56,57]. In this method, the polymer, such as PLA, is dissolved in a water-miscible solvent of transitional polarity, causing nanosphere precipitation. This occurs due to interactions with surfactants in stirred aqueous solution. The fast diffusion of the solvent forms a polymer deposition on the interface between water and the organic solvent resulting in the formation of colloidal dispersion [58]. A small volume of non-toxic oil can be incorporated in the organic phase for synthesizing nanocapsules, as well through a solvent displacement technique resulting in increased loading efficiency. This simple technique can only be used for water-miscible solvents to achieve a sufficient rate for spontaneous emulsification [58]. It is mostly applicable to lipophilic moieties because entrapment efficiencies as high as 98% can be achieved [59]. While it is relatively simple and cost effective, is not a first choice to encapsulate water-soluble drugs, such as DNA and RNA.

#### 3.1.3. Salting Out

This technique is based on the simple principle of separating a water miscible solvent from aqueous solution via salting out (Figure 5). The procedure is the improved and modified version of the emulsification/solvent diffusion. Here, an organic solvent, such as acetone, is used to dissolve the polymer and the drug, followed by its emulsification in an aqueous gel containing the salting-out agent (such as magnesium chloride, calcium chloride, or magnesium acetate), or non-electrolytes (such as sucrose) and a colloidal stabilizer (such as polyvinylpyrrolidone (PVP) or hydroxyethylcellulose) [60]. The formation of nanospheres is triggered by diluting the oil/water emulsion using an adequate amount of aqueous phase [61]. The salting-out agent plays a vital role in deciding the encapsulation efficiency of the drug; therefore, careful selection should be applied. Like the nanoprecipitation method, salting out also has a greater affinity to encapsulate lipophilic moieties [62].

#### 3.1.4. Dialysis

Dialysis is one of the simplest and effective methods to prepare small and uniformly distributed PNPs [54,63,64,65]. The method involves keeping the polymer (dissolved in an organic solvent) inside a dialysis tube with appropriate molecular weight range and performing dialysis against a non-miscible solvent. The homogenous suspension of NPs is formed resulting from solvent displacement inside the membrane followed by polymer aggregation due to a loss of solubility (Figure 6) [66]. The mechanism of PNP formation via dialysis method is yet to be understood. There are various polymers and co-polymers that have been synthesized using this technique [66,67,68,69,70,71] such as poly(benzyl-l-glutamate)-b-poly(ethylene oxide) and poly(lactide)-b-poly(ethylene oxide) NPs [72].

#### 3.1.5. Supercritical Fluid Technology

The supercritical fluid technology involves the use of supercritical fluids, which are more environmentally friendly as compared to conventional solvents, and have the potential to produce PNPs of high purity (Figure 7) [73,74,75]. This technology is based on two principles: (1) rapid expansion of supercritical solution (RESS) and (2) rapid expansion of supercritical solution into liquid solvent (RESOLV) [74].

Using the conventional RESS principle for supercritical fluid technology, well-dispersed particles are formed by dissolving the solute in a supercritical fluid, followed by the rapid expansion of the solution across an orifice into surrounding air. The high degree of super saturation along with prompt reduction in the pressure for expansion, results in the formation of homogenous particles [50]. Poly (perfluoropolyetherdiamide) droplets are produced using this technique. Keshavarz et al. were able to use this technique to prepare raloxifene NPs with the smallest particle size being 18.93 ± 3.73 nm and having a PDI less than 0.1 [76].

The modified version of the RESS method, which consists of the expansion of the supercritical solution into a liquid solvent as opposed to surrounding air, is called RESOLV (Figure 8) [73]. The RESS method produces a poorly separable mixture of nanometer and micrometer sized particles, with micro-particles being the primary product. Low production of nanometer size range particles through RESS was overcome by introducing the RESOLV technique [77]. In the RESOLV method, the liquid solvent makes it possible to achieve nanosized particles by suppressing the particle growth in the liquid solvent [73]. Meziani et al. reported the preparation of poly(heptadeca-fluorodecylacrylate) (PHDFDA) NPs using this technique with an *average* particle size of less than 50 nm [77].

Ultimately, while we include these methods in the review for historical perspective, top-down technologies are mostly preferred for the encapsulation of small molecules, often applicable for lipophilic moieties.

### 3.2. Bottom-Up Strategies for the Preparation of Polymer Nanoparticles

#### 3.2.1. Emulsion Polymerization

The method can be classified in two approaches depending upon the usage of the organic or aqueous continuous phase [50]. The continuous organic phase methodology involves dispersing the monomer into an emulsion or into a non-solvent material (Figure 9) [53]. However, the method demands for toxic organic solvents, surfactants, monomers, and an initiator, which are eventually washed off from the finally formed particles. The particles synthesized using this method are: poly (methylmethacrylate) (PMMA), poly(ethylcyanoacrylate) (PECA), and poly(butylcyanoacrylate) (PBCA) NPs, produced via surfactant-based dispersion into solvents such as cyclohexane (ICH, class 2), n-pentane (ICH, class 3), or toluene (ICH, class 2) as the organic phase [51].

The initiation is not required when the monomer is dissolved in an aqueous continuous phase. There are various other methods of inducing initiation such as high-energy radiation like gamma rays, ultraviolet (UV), or strong visible light. Mini-emulsion polymerization involves cocktails of monomers, water, co-stabilizer, surfactants, and initiator similar to emulsion polymerization. The factors that distinguish these two methods are the usage of a low molecular mass compound as a co-stabilizer, and the use of high-shear devices such as ultrasound generators. Mini-emulsions are disparagingly stabilized, calling for high-shear to achieve a steady state and have a high interfacial tension [73]. On the contrary, micro-emulsion polymerization results in having considerably smaller particle size and average number of chains per particle [50]. In micro-emulsion polymerization, a water-soluble agent acting as an initiator is mixed in the aqueous phase of thermodynamically stable micro-emulsion containing swollen micelles. The type and concentration of the initiator, nature of the surfactant and the monomer, and reaction temperature are a few factors influencing micro-emulsion polymerization kinetics and the properties of PNP [54,78].

#### 3.2.2. Recombinant Technology

Cationic polymers synthesized by utilizing recombinant DNA technology have the potential to address some of the major challenges of gene delivery such as the low ability to target cells, poor intracellular trafficking of the genetic material, and nuclear uptake. Synthetic methods of polymer production involving conventional thermodynamically-driven chemical techniques are inadequate for gene delivery purposes as the resultant products are heterogeneous with regard to composition and molecular weight. In contrast, amino acid-based polymers synthesized via recombinant technology in living systems, such as *E. coli*, produces homogenous biopolymers with a specific composition where function can be influenced by the amino acid sequence [79]. This paves the path for multiple functionality approaches, allowing for a single biopolymer to have multiple functions merely by changing the protein expression. Aris et al. reported the engineering of a gene delivery system, namely 249AL, composed of a cationic lysine oligomer (K_10_) conjugated with ß-galactosidase-derived protein displaying an arginylglycylaspartic acid (RGD) cell attachment peptide [80]. The K_10_ aids in the condensation of plasmid DNA (pDNA), whereas RGD interacts with the αVβ3 integrin present on the cell membrane [81]. The efficiency of the system was determined by complexing 249AL with pDNA encoding a luciferase reporter gene and transfecting CaCo_2_ cells. Since the 249AL was aimed to be target-specific, the percentage of transfected cells along with the total gene expression could help better understand the efficiency of the system. The transfection efficiency of 249AL was significantly lower compared to commercially available transfecting agents since 249AL was unable to perform an endosomal escape [81]. Similarly, Furgeson’s group reported the development of a recombinant elastin-based cationic di-block biopolymer for gene delivery [82]. The biopolymer consisted of a cationic oligomer block (VGK8G) conjugated with a thermo-responsive elastin-like polymer with 60 repeats of Val–Pro–Gly–Xaa–Gly (VPGXG). This particular approach is pseudo-biosynthetic, utilizing a recursive directional ligation method to synthesize the gene [82,83]. Similar to the 249AL, the biopolymer was not capable of escaping the endosome. To overcome this challenge, another group, Hatefi et al., reported the first recombinant cationic biopolymer with tandem repeating units of basic amino acids such as lysine (K) and histidine (H) residues conjugated with fibroblast growth factor 2 (FGF2) [84]. The biopolymer denoted as dKH-FGF2, with 36 lysine residues and 24 histidine residues, facilitated in condensing the pDNA and performing an endosomal escape via a proton sponge effect respectively [85]. FGF2 conjugation provided specific targeting to fibroblast growth factor receptor (FGFR) on cells such as T47D (breast cancer) and NIH3T3 (fibroblasts). The results showed that the biopolymer was able to condense pDNA into NPs and induced significant cell proliferation. While the result of the transfection efficiency studies suggested targeted gene transfer via FGFR, the biopolymer efficiency was not optimal, but showed potential for optimization [84].

Overall, the preparation of PNPs is yet to be perfected. Lipophilic drug loaded nanospheres or nanocapsules can be synthesized using simple, safe methods with good reproducibility. However, as we move to more complex and sensitive therapeutic cargos like genetic materials, careful selection of the appropriate method is crucial in order to achieve appropriate physicochemical characteristics (Table 2), minimal interference of process parameters and maximum entrapment efficiency.

### 3.3. Polymerization Chemistries for Common Synthetic Polymers

#### 3.3.1. Poly(Lactic Acid)

Some synthetic polymers can be designed via multiple types of polymerization chemistry. An example of such a polymer is PLA. It has been researched widely for gene therapy over the years due to its ease of availability, reasonable pricing, and biodegradable nature. The building block of this polymer is lactic acid, produced naturally via fermentation. The literature identifies two polymerization mechanisms for synthesizing PLA [86].

Direct condensation of lactic acid: It is the conventional method of synthesis utilizing solvents and exhibiting high reaction times [86]. It has been done using diphenyl ether as a solvent in the presence of tin (II) chloride as the catalyst. The process is strictly dependent on the polymerization temperature and pressure. An increase in temperature leads to a high molecular weight PLA [87]. Other solvent systems like p-xylene [88] have also been employed. Solid-state direct poly-condensation without the use of a solvent was proposed utilizing this reaction chemistry. A pre-polymer product was formed first, using p-toluene sulfonic acid without the addition of any catalyst. This product was then subjected to solid-state polymerization under high temperature and pressure conditions [89].Ring opening polymerization of lactide: This process is completed in two steps. In the first step, lactic acid cyclizes into lactide (a close chain lactone di-ester) under heat and a vacuum. A nitrogen-controlled inert environment is used to speed up the removal of water vapors, enhancing cyclization. The second step involves disruption of the cyclic ring, followed by the union of open chains forming the polymer. This step is catalyzed by stannous octoate to promote formation of ester linkages [86,90]. Process parameters and solvents used in this approach fulfills the requirements of “green chemistry” [90], a novel advancement in polymer nanoscience, which was discussed previously.

#### 3.3.2. Poly-l-Lysine

Poly-l-lysine (PLL) has been synthesized in many structural conformations and molecular weights, i.e., linear, dendritic, and hyper-branched, each exhibiting a characteristic safety profile [91]. Linear PLL is synthesized via polymerizing the monomers under heat and vacuum, followed by precipitation using diethyl ether. Dendritic PLL is created by conjugating lysine monomers as a branch unit. They have shown improved gene transfection compared to their linear counterparts [92]. Dendritic conformation utilizes an initiator core like hexa-methylene-diamine, on which lysine monomers are coupled repeatedly in various conformations [93]. Hyper-branched PLL is a relatively newer conformation structurally related to dendritic PLLs. Unlike dendritic conformations that exhibits a single core, hyper-branched PLL possesses a randomly branched structure, and it is attractive as it can be produced in a one-step operation [94]. No initiator core is involved and lysine monomers are polymerized using heat in an inert environment created using a nitrogen gas influx. Once polymerization is completed, the final product is collected via precipitation [94,95].

#### 3.3.3. Poly(Amidoamine)

Poly(amidoamine) (PAMAM) is also an attractive non-viral vector for gene delivery, especially for complex targets like cochlea in the inner ear [96] or glioblastoma in the brain [97]. It consists of an alkyl-amine core and tertiary amine branches in dendritic conformations. In most cases, the core utilizes ethylene-diamine-tetra-acetic acid (EDTA) [98].

#### 3.3.4. Poly(Methyl-Methacrylate)

Poly(methyl-methacrylate) PMMA is one of the extensively investigated polymers for its application in electrospinning [99], synthesis of carbon nanotube/PMMA composites, and for high refractive index thin-film fabrication [100]. The properties of the PMMA depends greatly on the resulting molecular weight. A bottom up approach of its polymerization involves ionic and free radical polymerization. The anionic polymerization consist of an active anionic center. It is a dynamic polymerization technique in which the chain termination does not occur until the addition of a terminating agent [101]. The degree of polymerization is determined by the molar ratio of monomer versus initiator, in the absence of a terminating agent. Anionic polymerization is used to produce PMMA-PS with a low molecular weight. The bottleneck of the anionic polymerization is the requirement for stringent reaction conditions as the anion is sensitive to an environment consisting of both oxygen and water. Hence, the process requires purification of all the polymerization reagents and inert atmospheric conditions [102,103]. The polymerization technique is used to synthesize high molecular weight PMMA and its related block copolymers. Another polymerization technique is reversible addition fragmentation chain transfer (RAFT). The technique is very similar to other free radical polymerization. In RAFT, the thermochemical initiator or the interaction of gamma or UV radiation with some reagents gives away free radicals [104]. High molecular weight PMMA can also be successfully synthesized by other techniques like activators regenerated by electron transfer (ARGET) or atom transfer radical polymerization (ATRP). These new techniques require a significantly smaller amount of Cu (II) species [105]. Conclusively, a number of bottom-up techniques are available for the synthesis of high molecular weight PMMA.

#### 3.3.5. Poly(Ethylene-Imine)

Twenty-five kilodalton (kDa) branched poly(ethylene-imine) (PEI) and 22 kDa linear PEI are the most common types employed for gene therapy [106]. The mechanism put forward for the synthesis of linear PEI involves a cationic ring opening polymerization of 2-oxazoline [106,107]. When acylated oxazoline is used, i.e., methyl or ethyl oxazoline, the reaction is processed via hydrolysis under strong acid and elevated temperature in the aqueous medium [107]. Like linear PEIs, branched PEI is prepared via a cationic ring opening polymerization of aziridine [106]. This is achieved through an electrophilic attack of protons on the aziridine monomer [106,108]. The overall polymerization reaction can be processed via the catalytic activity of acid and input of heat in aqueous or alcoholic solution [108]. Detailed reaction chemistry and the control of individual steps is presented by Jager et al [106].

PEI has been successfully developed as a commercial DNA transfection reagent. The system called jetPEI^TM^ produces an efficient gene transfection for up to 4 h with minimal cytotoxicity [109]. It is made of linear PEI chains and is useful for both adherent and suspension cells. Cell-specific versions are also designed for this reagent like jetPEI^®^-Macrophage (primary macrophages, glial, and dendritic cells), jetPEI^®^-Hepatocyte (liver cells), and jetPEI^®^-HUVEC (endothelial cells) [109].

#### 3.3.6. Poly(Lactic-co-Glycolide)

Poly(lactic-co-glycolide) (PLGA) is a synthetic copolymer of lactic acid and glycolic acid. PLGA can be synthesized via direct poly-condensation of lactic and glycolic acid; however, the most efficient and prevalent scheme to obtain high molecular weight copolymers is the ring opening polymerization of lactide and glycolide. The synthesis of high molecular weight PGLA using poly-condensation demands for a high degree of dehydration, which is difficult to achieve and is considered an inefficient method to obtain a good yield of polymers [110]. To prepare high molecular weight PLGA in a shorter reaction time, it is important to proceed via ring-opening polymerization of cyclic diesters, lactide, and glycolide.

Both PLA and PLGA are FDA-approved polymers used in commercially available drug products. They have been extensively used in drug delivery for reducing the lowest effective dose, targeting drug activity, and reducing the drug toxicity. To date, FDA has approved 15 drug products utilizing PLA or PLGA [111]. Some examples include:
PLGA containing drugs
i)Vivitrol (naltrexone) intramuscular (IM)ii)Zoladex (gorserelin acetate) subcutaneous (SC)iii)Lupron depot, Lupron (leuprolide acetate) IM, and Lupaneta pack (leuprolide acetate and norethindrone) oral and IMiv)Sandostatin LAR (octreotide) SCv)Trelstar (triptorelin pamoate) IMvi)Arestin (minocycline HCL) periodontalvii)Risperidal Consta (risperidone) IMviii)Ozurdex (dexamethasone) SCix)Bydureon (exanatide) tablets oralx)Signifor LAR (pasireotide pamoate) IM
PLA containing drugs
i)Lurpon depotii)Atridox (doxycycline) periodontal



Since their use is successful in the area of drug delivery, PLA and PLGA-based gene carriers are promising for achieving similar success for gene therapeutics.

## 4. Challenges Associated with the Use of Polymers in Nanomedicine

Every new technology is a two-edged sword. Like many other non-viral vectors, PNPs also have some challenges in terms of safety and stability as nanocarriers. As new generations of PNPs pave their way toward clinical trials as vectors, focus is laid on overcoming these challenges. PNPs with optimum shape and flexibility for the best interaction with the cellular membrane, and which exhibit compartmentalization for loading genes and targeting moieties, are designed. To reduce the systemic toxicity, PNPs that mimic biological molecules are created. Importance is given to reduce trafficking and off-target accumulation, and finally, effort is made to introduce stimulus-responsive behaviors. The following paragraphs will briefly discuss the limitations of PNPs and advancement in research to overcome them (Figure 10).

### 4.1. Stability of PNPs in an Electrolyte and Protein-Rich Biological Medium

Polyplexes formed between cationic polymers and genetic materials are based on electrostatic interactions. Factors including the molecular weight, hydrophilicity, surface charge, and structure of the cationic polymers define the efficacy of the carrier [15]. Among common PNPs, PLA (degradable) and PMMA (non-degradable) are most noteworthy as they are approved for human use by FDA [112,113]. Lazzari et al. analyzed the colloidal stability of these polymers in simulated biological media based on different characteristics. PLA NPs showed a high degree of aggregation and a 20% increase in the mean particle size distribution in synthetic gastric juice, while PMMA NPs remained stable. This was anticipated due to the difference in the surface charge densities (i.e., zeta potentials) of the two formulations. If the NPs aggregate during circulation, the risk of clotting increases and overall release of the drug is compromised. PMMA NPs showed higher stability in serum and other organ homogenates as compared to PLA NPs due to higher surface charges [114].

Another prominent parameter controlling PNPs stability in biological fluids is the hydrophobicity of the surface. For example, a less hydrophobic version of *N*-isopropyl-acrylamide and *N*-tertiary-butyl-acrylamide copolymer particles hindered protein binding on the surface as compared to a more hydrophobic version, which bound series of proteins including apo-lipoproteins, albumin, and fibrinogen, thus increasing the risk of phagocytic destruction [115]. Unlike many other NPs, the challenge for PNPs is that changing the hydrophobicity of polymers is very difficult without adversely affecting the size, surface charge, and composition [116].

Despite the instability of some NPs in serum rich media, few polymers when conjugated with other organic and inorganic NPs improves the overall stability of the carrier and reduces opsonization during blood circulation. A typical example put into practice for many NP formulations is PEGylation. Attachment of PEG on the surface extends the circulation time of NPs by effectively reducing random interactions with proteins, recognition by immune cells, and clearance through excretory organs [117,118,119].

### 4.2. Accumulation and Toxicity of Polymeric Nanoparticles

PNPs made of non-degradable polymers tend to accumulate in the organs, mainly the liver and spleen, leading to toxicity [120]. The most studied example for drug delivery purposes is dextran, a glucose-based polymer [121,122,123,124]. Serious allergic reactions including anaphylaxis, volume overload, pulmonary and cerebral edema, and platelet dysfunction have been reported via in vivo studies on this carrier [123,124,125]. Dextran also exhibits a strong osmotic effect leading to acute renal failure and hence its use is contraindicated in patients with renal insufficiency and diabetes mellitus [126]. Although, the knowledge gained for this system might not be directly applicable to synthetic polymers used now for gene delivery, the overall chemical composition, particle size, shape, and surface properties of PNPs are contributors to their safety profile.

PLL is one of the primary cationic polymer used as a non-viral gene vector [127]. Unmodified PLL shows high in vitro toxicity, reducing the cell viability to as low as 50% [128]. Multiple efforts have been put forward to improve the biocompatibility of this carrier. Coating of PLL with PEG ranging from 5–25 mol% improved the transfection and biocompatibility for up to 96 h [128]. In recent years, more sophisticated coatings have been performed to further improve both in vitro and in vivo biocompatibility. An example of such a design is PEG-PLL-PLGA copolymers, which showed no blood toxicity or genotoxicity in vivo and revealed a wide safe scale of dosing [129].

Polyethylenimine (PEI) is another pioneer polymer used for gene delivery. It is the proof-of-concept for the design of NPs capable of condensing genetic materials and allowing intracellular endosomal escape to enhance transfection efficiency [130]. This cationic polymer has also shown considerable toxicity, depending on the structure (either linear or branched) and molecular weight [131,132]. Low molecular weight and moderately branched design is generally considered safer as compared to linear PEI [132]. However, the literature shows evidence that branched PEI has lower IC50 values as compared to linear PEI (37 mg vs. 74 mg, respectively) [133]. Moreover, the number of primary amines on the surface, and hence the surface cationic character, is directly proportional to hematological compatibility and cytotoxicity [133]. In free form, PEI may interact with the negatively charged serum proteins and red blood cells, destabilizing the plasma membrane and leading to immediate toxicity [134]. As complexes with genetic materials, this effect decreases but can be restored once the cargo is transported and free PEI appears back in the blood for excretion. Free PEI also interacts with cellular components and interferes with the normal processes [135,136]. PEGylation of PEI NPs is one the approaches presented to improve their safety profile. Conjugation of PEG at 6 and 10% weight ratios significantly reduced the cytotoxicity and aided in the local delivery of genes to the muscles. This study showed that a higher PEG ratio on the surface exhibits better safety [137]. Another approach tried to reduce the toxicity of free PEI is the coating of PEI/DNA polyplexes with lysinylated, histidylated, and arginylated cholesterol. This morphology improved the nuclear delivery of DNA at low PEI/DNA weight ratio of 2, reducing the need of excess free PEI chains in the system [138]. Only a few approaches are defined here; however, over time many designs have been put forward to minimize and even mitigate the safety challenges for these two most common PNPs.

Novel type of PNPs are dendrimers, which are repeatedly branched large spherical structures having a core, an inner shell, and an outer shell [139]. PAMAM is one of such well-known dendrimer used as chemotherapeutic and non-viral DNA delivery vehicle [140,141,142]. Amine-terminated positively charged PAMAM dendrimers exhibited hemolysis in rat blood cells by destabilizing the cell membrane [143]. Due to excellent delivery capabilities of this carrier, many approaches were assessed to overcome this safety challenge. One such approach is to change the type of the terminal amine functionalities. Replacing primary amines with secondary or tertiary amines reduced the cytotoxicity of this carrier [144]. Fatty acid functionalization or PEGylation (discussed at length in the following chapter) of PAMAM dendrimers reduced the cytotoxicity of the whole conjugate [145]. It is also believed that hydroxyl or meth-oxy terminated dendrimers are relatively safer than other counterparts [125].

Physiological behavior of many PNPs also depends greatly on the encapsulated payload, as this parameter changes its innate size, charge, stability, and other physicochemical properties. For example PEG-PLGA NPs encapsulating a trial drug “KU50019” at various ratios yielded different particle sizes. The smaller drug-encapsulated PNPs showed higher small bowel toxicity than larger particles [146].

### 4.3. Oxidative Degradation of Polymers—Generation of Toxic Metabolites

Common biodegradable polymers are generally regarded as safe, but there is a scarcity of data collected on the short-term and long-term toxicity resulting from the accumulation of polymer degradation products in the body. One elaborative study investigated in vitro toxicity of some polymers including PLGA, poly-ε-caprolactone (PCL), and poly(lactide-co-caprolactone) (PLCL), by studying the mechanisms of cytotoxicity induced through such metabolites [147]. The study concluded that all the above polymers exhibited concentration- and time-dependent damage to healthy cells including macrophages, hepatocytes, lung and kidney epithelial cells, and neuronal cells [147]. It is debatable whether these results can be extrapolated to in vivo scenarios as the human body is able to eliminate such metabolites through various pathways (e.g., lactate may undergo the Krebs cycle), which in turn retains the biocompatibility of the overall system [148,149,150]. However, the possibility of toxicity based on concentrations and time of exposure persists if these degradative products are not readily cleared out from the site of accumulation.

PLGA is the most widely accepted FDA-approved biodegradable polymer [125,151,152]. Although degradation products of PLGA are easily metabolized and eliminated by the body, and therefore the systemic toxicity is limited, it is highly dependent on the ratio of lactide/glycolide components and the total molecular weight [125,151,153]. Toxicity assays for PLGA NPs performed in mouse models showed no tissue damage, but exhibited 40% retention in the liver following oral administration [154]. Most studies conclude that, although PLGA NPs may tend to accumulate in tissues to a certain degree, they show very limited cytotoxicity or inflammatory responses irrespective of their surface properties. However, many times the use of PLGA in gene therapy requires further functionalization to improve gene transfection. This can lead to changes in their innate safety profile. So far, the functionalization of PLGA NPs with other active molecules including secondary polymers have shown to either have no effect or compliment the innate biocompatibility. Thus, no studies in the literature provide any evidence that functionalization may trigger loss of innate PLGA safety. For instance, PEGylation of PLGA NPs are shown to improve sub-cellular organelle targeted delivery, minimizing the random exposure and accumulation of polymer complexes thus complimenting its safety [155]. On the other hand, functionalizing PLGA NPs with poly(ε-carbo-benzoxy-l-lysine) to improve the loading capacity of vascular endothelial growth factor did not have any effect on the polymer’s innate biocompatibility [156].

### 4.4. High Cost Associated with Biological Analyses and Process Development

In addition to the above-mentioned challenges, another major concern that seems to be common for all nanoformulations is the time and cost associated with the assessment of their safety and efficacy. Since newer polymers are self-destructing biodegradable polymers, it is crucial to elucidate their in vivo metabolism and elimination routes. This further adds to the efforts and resources required for the formulation, development, and metabolite tracking. Therefore, although this area has consistently shown value for drug/gene delivery, many designs do not successfully reach the stage of clinical translation. Moreover, bench-top research now focuses on multifuntional PNPs containing a combination of functional groups to achieve encapsulation, delivery, and targeting. More functionalization means more sophisticated reaction chemistries, a need for elaborative tracking for degradation products, and more stringent process design parameters, which are all associated with higher cost, development times, and regulatory hurdles. Moreover, when these formulations are introduced in the marketplace, they are generally very expensive. Launching generic counterparts quickly after approval and bringing government and private drug plans on board for these technologies would make them more accessible to the general patient population.

## 5. Biodistribution and Cellular Interaction of Polymeric Nanoparticles

PNPs are capable of overcoming multiple biological barriers and controlling the release of their therapeutic load. However, preventing a rapid clearance of circulating NPs is an acute issue for their application to full potential; therefore, it is imperative to understand the factors influencing their circulation time and biodistribution. These factors define the ability to overcome body’s defense mechanisms and include tailorable physiochemical properties, such as composition, configuration, size, core properties, and surface functionalization (PEGylation, charged moieties, and targeting ligand functionalization) [157]. The interaction of the NPs with the biological environment is mainly influenced by opsonization, where opsonin proteins found in the blood serum interact and expose the NPs to macrophages in the mononuclear phagocytic system (MPS), triggering their removal from the biological system [120]. After opsonization, phagocytosis occurs, which involves the engulfing and destruction of foreign material from the bloodstream. PNPs that cannot be eliminated by phagocytes are sequestered in the MPS organs (liver and spleen). Here it is important to note that in the case of non-biodegradable PNPs, accumulation in these target organs may result in negative outcomes [158,159,160]. Overall, the process of in vivo trafficking in the blood stream compromise their designed therapeutic function.

There are no straight-forward strategies to completely avoid opsonization of NPs. However, significant research in this area over the last 35 years have identified patterns and method for increasing the overall blood circulation half-life and effectiveness of the delivery system. For example, research provides evidence that opsonization of hydrophobic particles is quicker compared to hydrophilic counterparts due to the enhanced adsorption of serum proteins on those surfaces [161,162,163]. For hydrophilic NPs, in vitro studies have established a correlation between surface charge and opsonization, indicating that neutrally charged particles have a much lower opsonization rate than charged particles [164]. There are several polymeric systems that have been used as shielding groups to reduce protein adsorption, such as polysaccharides, polyacrylamide, poly(vinyl alcohol), poly(Nvinyl-2-pyrrolidone), PEG, and PEG-containing copolymers, such as poloxamers, poloxamines, and polysorbates. Out of all the polymers tested so far, the most effective solution is PEG or PEG-containing copolymers [120], thus we discuss the details of its biological profile and current developments in the next segment.

### 5.1. Biodistribution of the PEGylated Polymers

The PEGylated stealth improves the residence times in the blood and ultimately the efficiency of PNPs. However, the rate of clearance and final biodistribution in the organ is dependent on multiple factors. First and foremost, the particle size plays a crucial role in defining the final distribution and blood clearance of the PEGylated PNPs. Particles with a hydrodynamic size over 200 nm typically exhibit a more rapid rate of clearance than particles under 200 nm, regardless of PEGylation [165]. In addition to the blood clearance rate, the final biodistribution is also affected by particle size. For instance, PEGylated NPs sized less than 150 nm was shown to exhibit a higher uptake in the bone marrow of rabbits, whereas particles sized 250 nm mostly sequestered in the spleen and liver, with only a small fraction of uptake by the bone marrow [165]. Moghimi et al. [166] suggested that the size-dependent biodistribution might be attributed to a filtering effect, whereby the spleen and liver removes larger particles more rapidly, while the smaller particles are directed to the bone marrow. However, the exact reason for these size dependencies is yet to be understood. Like particle size, molecular weight also plays an important role. Molecules that have a molecular weight less than 5000 kDa, or even more conservative for dense polymers like dendrimers, can be removed from the body via the renal system [167].

Due to the hydrophilic nature and capability to modulate NP size, PEGylation has been shown to reduce liver accumulation by half to one-third [167,168]. This feature is attributed to the stealth effect, i.e., diminished random protein binding during circulation [168]. Moreover, macrophages and liver Kupffer cells, which are the major limiting factors, hinder the circulation times of NPs. Hydrophilicity of PEG on the NP surface reduces its tendency to attach opsonin proteins, thus reducing their recognition and transport to the liver for excretion via these cells [120]. One such example is the PEGylation of gadolinium oxide, a fluorescent NP utilized for both imaging and phototherapy [169]. Gadolinium oxide NPs embedded in a polysiloxane shell was functionalized with PEG monomers with different chain lengths and terminal groups. These include PEG250-COOH, PEG2000-COOH, PEG2000-NH_2_, and PEG2000-OCH_3_. Three PEGs containing amino or carboxylic groups at both ends, and one PEG containing methoxy group on one end and carboxylic group on the other end, were covalently grafted on the surface of these fluorescent hybrid NPs. Complexes grafted with PEG2000-COOH and PEG2000-OCH_3_ prevented the accumulation in the liver, spleen, and lungs. PEG2000-NH_2_ localized these particles largely in the liver and spleen. Moreover, shorter chained PEG failed to localize NPs at the tumor site [169].

Jones et al. [170] studied the delivery of siRNA via NPs of triblock copolymers consisting of hyper-branched PEI grafted-PCL-block-poly(ethylene glycol) (hy-PEI-g-PCL-b-PEG) with and without a folic acid targeting ligand [171,172,173]. To mimic the clinical manifestations of ovarian cancer, an animal model was created with cells that overexpress folate receptor α to an accuracy degree of 85%. The overall tumor uptake of targeted NPs was 3.4% compared to 2.4% for non-targeted particles. A strong uptake of the targeted formulations was observed 4 h post-injection in the primary tumor as well as metastases. It decreased over time due to NP washout. At 24 h post-injection, only diffuse uptake and targeting was observed [170].

There are two general patterns that seem to be consistent throughout most biodistribution studies. First, a larger molecular weight PEG coating on NPs results in longer blood circulation half-lives in most vivo studies [174]. The second common pattern exhibited is that uncoated NPs accumulated more in the liver, whereas PEGylation hindered this shift [159]. For example, 24 h after injection, 40% of PEGylated particles were found in the liver, while 90% of naked NPs resided in the liver after only 3 minutes of injection. On the other hand, PEGylated NPs were found to accumulate more readily in spleen. Over 60 min of blood circulation, the concentration of PEGylated particles in the spleen was 12% while it was only 2% for “naked” particles [174].

### 5.2. Translational Concerns Regarding PEGylation of Gene Complexes

Although PEGylation is a widely researched approach for the efficient delivery of drug molecules [119,175], only a few formulations have paved their way in clinical trials as marketed drugs [176]. Two major physicochemical challenges are identified with PEGylation of polymers: (1) the lack of conjugation sites and poor options for copolymerization, allowing only one drug molecule to be loaded per linear chain [177]; and (2) PEGylation may impact the pharmacodynamic behavior of drugs at the site of binding due to steric hindrance at the target binding site [178,179]. Formation of PEG-based dendrimers and block copolymers can overcome the first challenge and allow multiple drug loadings per chain [180].

Moreover, there is growing evidence challenging the gold standard position of PEGylation in nanomedicine, including PEGylation of gene-delivery PNPs [179]. Some translational challenges regarding the use PEGylated NPs in clinical settings are (also summarized in Table 3):As PEG-based dendrimers and block copolymers are designed, it is imperative to control the aggregation and micelle formation that can compromise the biocompatibility and in vivo stability of the formulation [180]. For instance, injectable products should have a particle size restricted to <0.5 µm with a maximum aggregation limit of 5 µm to avoid serious embolic episodes.Immunogenicity of PEG molecules is a concern with the identification of about 25% of patients being positive for anti-PEG antibodies in their blood [181]. Anti-PEG antibodies have been reported after the administration of PEG-uricase [182,183] and PEG-glucuronidase [184].For PEGylated proteins and peptides, it is considered that PEG may cause partial fragmentation of proteins giving rise to new epitopes, or the methoxyl terminus of the PEG molecule can cause antigenicity in the blood.Difficulty in characterizing PEGylated drugs also remains a concern [179].Remnants of reaction chemicals used in PEGylation can be a concern for sensitive drugs and biomolecules [185].Patients with renal or liver insufficiency may be exposed to an increased risk of toxicity. For example, 240 grams of PEG 400 exposed a patient who was concomitantly taking lorazepam to acute renal tubular necrosis; therefore, continuous drug monitoring is required for such patients who are receiving a cocktail of medications [179].

However, developments in polymer science and the approval of new PEGylated drugs give an impression that these challenges are being resolved and lessons are being learned from small molecules that could be applied to polymeric and other NP-mediated gene delivery. An example of such an illustration is the development of PEGylated liposomal irinotecan injection for metastatic pancreatic cancer [186]. PEGylation has improved the half-life and exposure profile of irinotecan, improving its efficacy and maximum tolerated dose. Moreover, it allowed for the dose reductions required for the therapeutic effect of the drug while minimizing serious side effects like diarrhea and neutropenia. PEGylation has also been translated to improve the half-life of biopharmaceuticals and gene therapeutic products. The PEGylation of lysine residues in metalloproteinases was successful in retaining the enzyme activity while increasing the plasma half-life from 1.1 h to 28 h [187]. Similarly, a commercial product of PEGylated antibody interferon α-2a (Pegasys^®^, Roche, Basel, Switzerland) shows improved pharmacokinetic performance, allowing weekly injections for hepatitis C as compared to its unmodified form [188]. To date, there is a scarcity of concrete assessments regarding the benefit/risk profile of PEGylated PNPs in the context of gene therapy, but some studies have been presented. PLL coated with PEG in different mole ratios, as described previously in the review, exhibited fast and sustained gene expression for up to 96 hours and improved cell viability [128]. Later, this model was complexed with a positively charged fusogenic peptide [189]. This modification minimized PEG driven aggregation of complexes and further improved gene transfection. Moreover, the cell viability was maintained close to 100% [189]. However, both studies based their conclusions merely on cell viability assays and failed to assess metabolic determinants and long-term safety and stability of the designs [128,189]. Therefore, it is imperative to elucidate oxidative, metabolic, and functional stresses at cellular and organ levels, and in vivo biodistribution, organ retention, and mode of elimination for successful translation of this gene carrier. In light of the current evidence, PEGylation remains one of the best options for protection and controlled release of genetic material from PNPs.

## 6. Non-Invasive Routes of Administration of PNPs

The most common way of administering non-viral gene delivery platforms into the living system is via parenteral route: intravenous, intramuscular, intraperitoneal, or subcutaneous injections [190]. These modes are effective and efficient; however, they come with challenges, such as the need for trained medical staff for administration, being invasive, reduced patient compliance, and can result in an undesired off-target effect. Capitalizing on the versatility of polymers, current research explores non-invasive and minimally invasive routes of administration (Figure 11) to overcome the shortcomings of the injectable gene therapy drugs.

### 6.1. PNPs for Oral Administration

Oral delivery presents a stimulating alternative approach to improve the ease of administration and promote patient compliance. It also provides access to a large cellular surface area (i.e., intestinal epithelium) for transfection. Furthermore, the ability to elicit local and systemic responses is one of the major theoretical advantages [192,193]. However, the route has its own challenges such as overcoming different conditions encountered along the GI tract including the variation in pH from 1 in the stomach to 7 in the small intestine and colon, or preventing the therapeutic agent from enzymatic degradation [194]. Additionally, the 450-µm thick intestinal mucus layer poses another obstacle [194]. Therefore, vigilant composition selection and design is mandatory to make a successful oral gene delivery system. This chapter will discuss the improvement of PNP stability within the GI tract, including enhanced particle uptake and increased cell targeting, to potentiate orally delivered gene therapies. Target specificity induced via functionalization of polyplexes is needed to improve the gastric stability of genetic material. For instance, Chunbai et al. designed mannose-modified tri-methyl-chitosan-cysteine NPs, which successfully encapsulated TNF-α siRNA for oral therapy against inflammation-related actuate hepatic injury [195]. These polyplexes were stable in both gastric and intestinal fluids.

Results from animal models of the above mentioned system used to orally induce gene therapy against acute hepatic injury demonstrated low TNF-α in the serum and low TNF-α mRNAs in spleen, lung, and liver macrophages [195]. Bile conjugates of functionalized NPs demonstrated enhanced gastric stability and particle uptake via intestinal enterocytes [196]. Nurunnabi et al. demonstrated the use of these PNPs as a way to treat type 2 diabetes. NPs of DNA and branched PEI (bPEI) were coated with taurocholic acid-conjugated heparin, improving the oral stability, as well as intestinal enterocyte uptake of these NPs. The treatment encoding glucagon like peptide-1, a stimulator of insulin secretion by pancreatic cells, were delivered orally to potentiate non-invasive treatment for type 2 diabetes. A normal blood glucose level was maintained for 21 days [196], suggesting a potential for translation of these NPs into clinical use.

Another strategy for overcoming gastrointestinal barriers is the use of dual material particulate systems capable of protecting the therapeutic payload during gastric transit and allowing its release in the intestine. Bhavsar et al. designed such a system for the treatment of inflammatory bowel disease. It was composed of solid, protective PCL microspheres coated over gelatin NP/DNA complexes. The system upon reaching the intestine gets preferentially degraded via enzymatic activity and releases the therapeutic DNA encoding anti-inflammatory cytokine interleukin-10 [197]. The PCL microsphere matrix provides an additional layer of protection to gelatin/DNA NPs. In a mouse model of acute colitis, the delivery system significantly reduced pro-inflammatory cytokines and disease severity [197]. A similar system was designed that encapsulated both siRNA and a pharmacological agent in a hyaluronic acid (HA)-functionalized, chitosan-coated (CS-coated) PLGA dual-material NPs. It was designed to alleviate inflammation associated with ulcerative colitis [198]. Oral delivery of the PLGA/CS/siRNA system created an anti-inflammatory environment within the gut, protecting the mucosal layer in a mouse model of ulcerative colitis [198].

The dual-material particulate system has also been investigated in cancer therapy. SiRNA/gold NPs encapsulated in CS-taurocholic acid were formulated to treat secondary hepatic cancer caused by metastatic colorectal cancer [198]. This dual-coated system, when delivered orally, reduced the expression of Akt2 proto oncogene in the liver, increased apoptosis of cancer cells, and reduced tumor size significantly more than the polyplexes made with gold NPs coated *only* with CS [199].

Based on these studies, PNP mediated DNA and siRNA delivery via the oral route holds a promising potential for local and systemic gene therapy.

### 6.2. PNPs for Topical Therapeutics

Stratum corneum, the top-most layer of the epidermis, is the rate-limiting step for topical therapeutics [200]. In order to transport any type of therapeutic moiety across this layer, it should be made such that it is capable of either penetrating via the intracellular route or cross via extracellular spaces through passive diffusion. However, diffusion is not always feasible for drugs with large molecular weight or hydrophilic moieties like pDNA, siRNA, or antibodies [201]. PNPs are widely researched for cutaneous drug delivery due to their flexible morphology. Most formulations developed in these studies were capable of delivering small molecules or highly lipophilic drugs via this route. Moreover, due to insufficient capabilities for overcoming cutaneous barriers, there is only a limited the number of FDA-approved drugs administered as transdermal preparations [202].

For cutaneous gene delivery, PNPs have only recently been considered. One such study employs chitosan, whereby multiple compositions of chitosan NP/DNA complexes were prepared via gelation using sodium tri-poly-phosphate. Particle sizes ranged from 200 to 287 nm, and a positive surface charge from 20.8 to 29.2 mV provided more than 90% encapsulation capacity. β-galactosidase expression was recorded following pDNA transfection. It was expressed continuously in NP-treated skin during 7 days of observation. A 4-fold higher transgene expression was recorded in baby rats as compared to adult rats. Throughout treatment, no skin irritation or redness was observed [203]. This study demonstrated the feasibility of cutaneious gene delivery and opens the way for optimization and expansion of PNPs in the area.

## 7. Non-Conventional Polymeric Designs for Gene Delivery

Polymeric hydrogels are relatively newer platforms used in gene delivery science. They are scaffolds of cross-linked polymers from natural and synthetic origins [204]. Although there have been many types of polymeric hydrogels developed over time [204,205,206,207], cyclodextrin (CD)-based hydrogels are one of such developments designed to improve many aspects of gene delivery like:-Achieving higher gene transfection as compared to conventional PNPs-Achieving sustained and controllable release of therapeutic genes-Achieving cellular targeting to improve localization of polyplexes within in vivo tumor models.

CD has been conjugated with cationic polymers like PEI and synthesized as a film to achieve the sustained delivery of genetic materials. PEI conjugated with CD formed polyplexes with pDNA, and this complex was embedded in poly-lactic acid films to achieve sustained gene release over the course of 28 h [208].

Polyrotaxane is a unique structure developed by combining linear polymers with CD. This structure is created through mechanical interlocking between straight polymeric chains like PEI, PEG, or poly(propylene glycol) and CD via non-covalent interactions, and terminated with bulky molecules providing unique properties like targeting [209]. One such design was synthesized using a PEG chain with alpha-CD and benzyl-oxy-carbonyl tyrosine as terminating groups [210]. The system complexed pDNA at a very low charge ratio of 0.5, yielding polyplexes with an optimum size (178–189 nm) and an overall positive charge (ζ-potential + 4.8 mV). The transfection efficiency improved significantly when compared to polyplexes formed with linear PEIs. Figure 12 illustrates the chemical structure of the complex and expected condensation and de-condensation for releasing pDNA intracellularly [210].

Polypseudorotaxane combining CD with a cationic block copolymer methoxy-poly(ethylene glycol)-b-poly(ε-caprolactone)-b-poly(ethylene imine) (MPEG–PCL–PEI) as the linear string is another example showing the application of a polymeric hydrogel for the sustained release of therapeutic genes over the course of 7 days. It was successful in achieving a controllable expression of BCL-2 conversion genes, downregulating this anti-apoptotic protein in a hepatocellular carcinoma model [211].

A CD-based polyrotaxamer is also used for combining gene delivery with cellular targeting for in vivo administration [212]. Here, PEI was linked to CD and folate molecules were used as terminating moieties for target localization of these NPs at the tumor site. These particles were able to form polyplexes at 2–3 charge ratio with an overall positive charge (+20–30 mV). In vitro assays revealed that the transfection was specific in the cells expressing a high number of folate receptors, while in vivo analyses also confirmed high tumor suppression in mice with this system when compared to other delivery controls. 

## 8. Conclusions

This review summarized the current standing of polymers in nanomedicine and highlighted the diversity of approaches for its design and utilization. Current evidence depicts that preparing PNPs is a dynamic state-of-the-art technology requiring thorough research for selecting a tailored synthetic approach. Moreover, there is still a need for analyses focused on improving the design for optimum biodistribution and therapeutic efficacy. Many factors regarding synthesis and functionalization, such as the application of green chemistry, reproducibility of process conditions, control of particle parameters like the droplet size, and overall surface charge, are yet to be fully understood. The review also summarizes the current understanding and encountered challenges regarding complex stealth technology through the PEGylation of other therapeutic moieties and NPs. With more research in this area and proper long-term characterizations, the clinical translation of many new designs is possible in the future. This work also summarized the applications of PNPs for non-invasive gene delivery over the years. A relatively simple and novel combinatorial designof both natural and synthetic polymers rendered PNPs suitable to deliver genetic material by many routes of administration. For instance, the oral delivery of polyplexes have not only targeted gene therapy for local issues like IBS but has been useful for dealing with systemic ailments like diabetes and hepatic injury. Development of topical scaffolds are now showing success in overcoming epidermal barriers to achieve systemic gene expression without being an irritant to the layers of the skin. Over time, more sophisticated combinational strategies have been proposed to simultaneously improve multiple aspects of gene delivery. Polymeric hydrogels like CD-based polyrotaxane and polypseudorotaxane have opened new avenues for working with practical in vivo targets for the controllable delivery and expression of therapeutic genes.

## Figures and Tables

**Figure 1 polymers-11-00745-f001:**
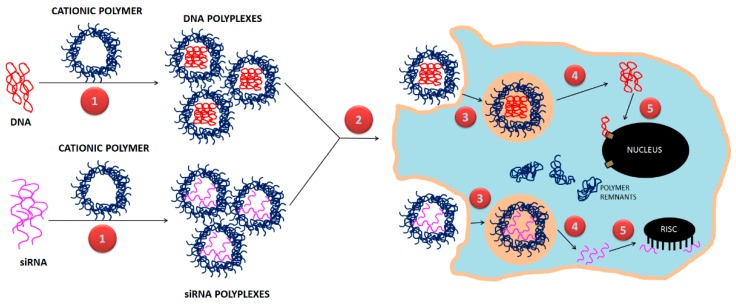
Polymeric nanoparticles for the intracellular delivery of DNA and siRNA: (1) complexation of anionic DNA and siRNA with cationic polymers to form polyplexes (2) cellular uptake of polyplexes via different endocytic routes, (3) enclosure and subsequent release of polyplexes from endo-lysosomal compartments, (4) release of free DNA and siRNA from polyplexes leaving behind polymer remnants, and (5) transfer of DNA to the nucleus for expression by nuclear membrane transport proteins and binding of siRNA by RNA-induced silencing complex (RISC).

**Figure 2 polymers-11-00745-f002:**
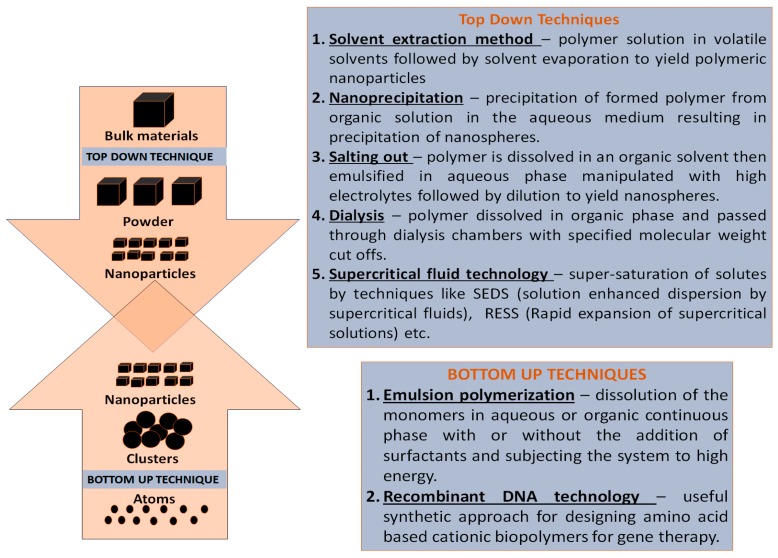
Summary of top-down and bottom-up techniques for generating polymeric nanoparticles.

**Figure 3 polymers-11-00745-f003:**
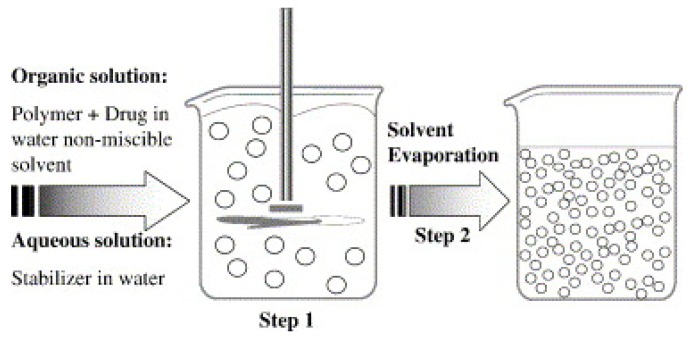
Schematic representation of the solvent-evaporation technique. Reprinted with permission from Reference [53]. Copyright 2006 Elsevier.

**Figure 4 polymers-11-00745-f004:**
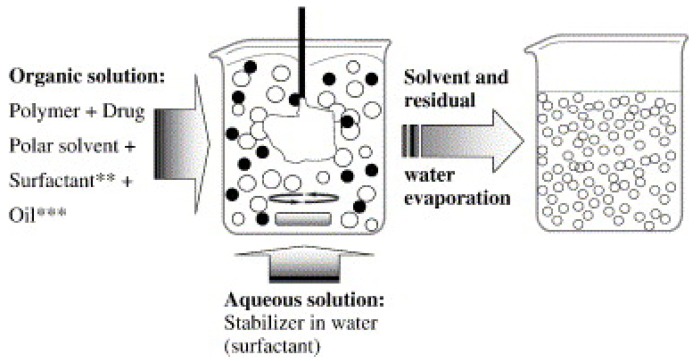
Schematic representation of the nanoprecipitation (solvent displacement) technique. Reprinted with permission from Reference [53]. Copyright 2006 Elsevier.

**Figure 5 polymers-11-00745-f005:**
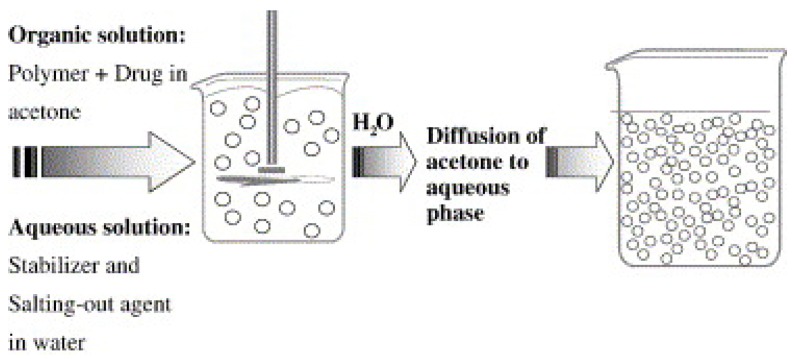
Schematic representation of the salting out technique. Reprinted with permission from Reference [53]. Copyright 2006 Elsevier.

**Figure 6 polymers-11-00745-f006:**
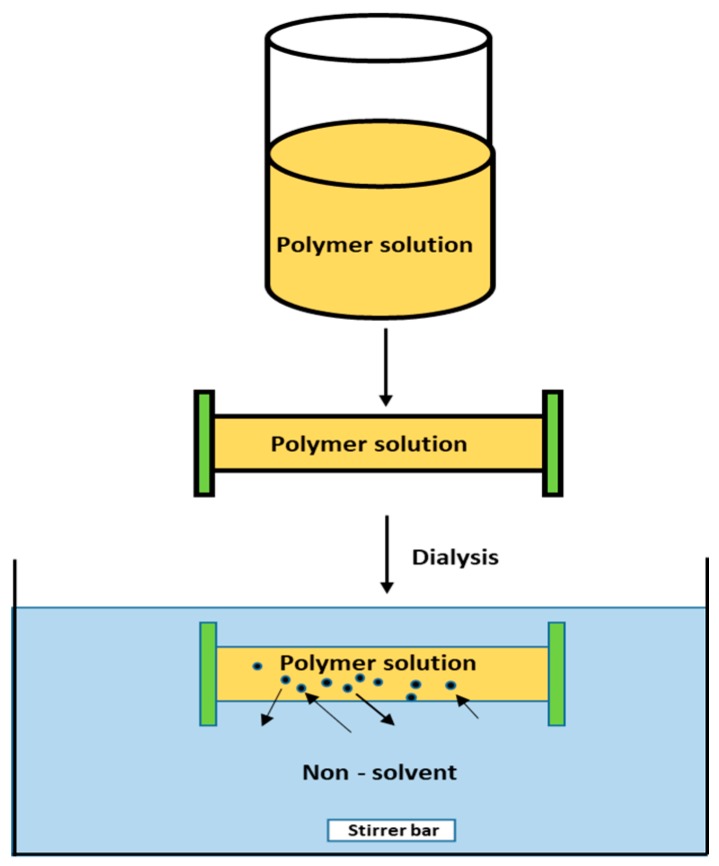
Schematic representation of an osmosis-based method for the preparation of polymer nanoparticles.

**Figure 7 polymers-11-00745-f007:**
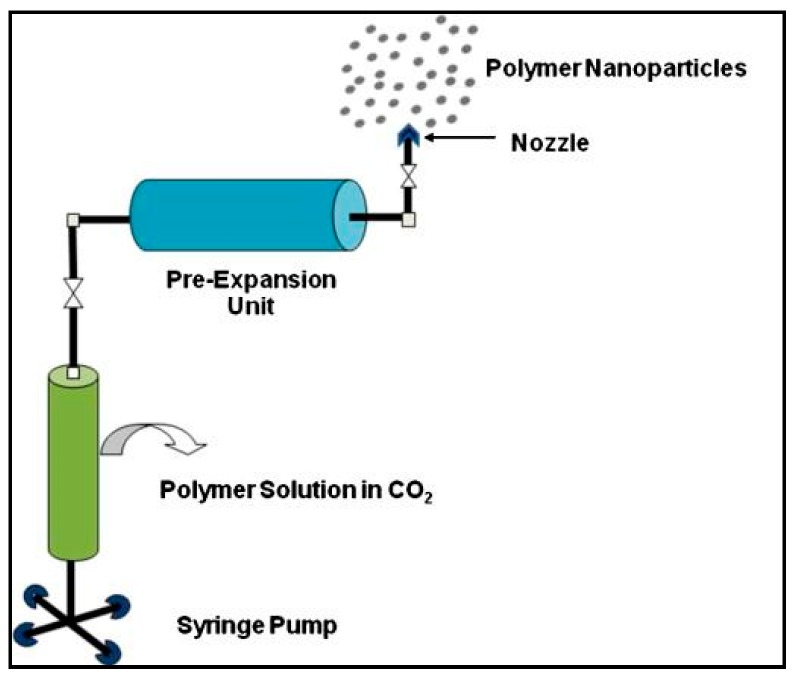
Experimental set-up for preparation of polymer nanoparticles via the rapid expansion of supercritical fluid solution. Reprinted with permission from Reference [73]. Copyright 2011 Elsevier.

**Figure 8 polymers-11-00745-f008:**
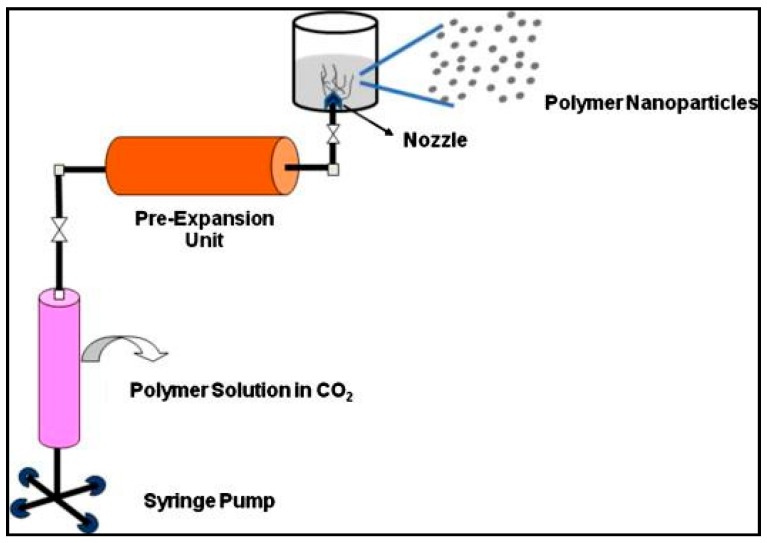
Experimental set-up for the rapid expansion of supercritical fluid solution into liquid solvent process. Reprinted with permission from Reference [73]. Copyright 2011 Elsevier.

**Figure 9 polymers-11-00745-f009:**
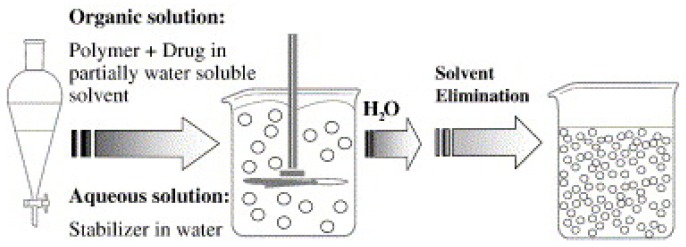
Schematic representation of the emulsification/solvent diffusion technique. Reprinted with permission from Reference [53]. Copyright 2006 Elsevier.

**Figure 10 polymers-11-00745-f010:**
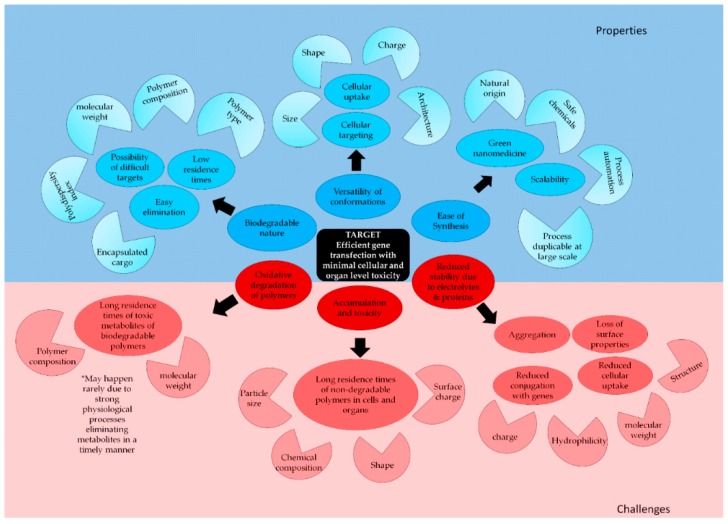
Summary of properties and challenges of polymeric nanoparticles for gene delivery and associated factors influencing each of these parameters.

**Figure 11 polymers-11-00745-f011:**
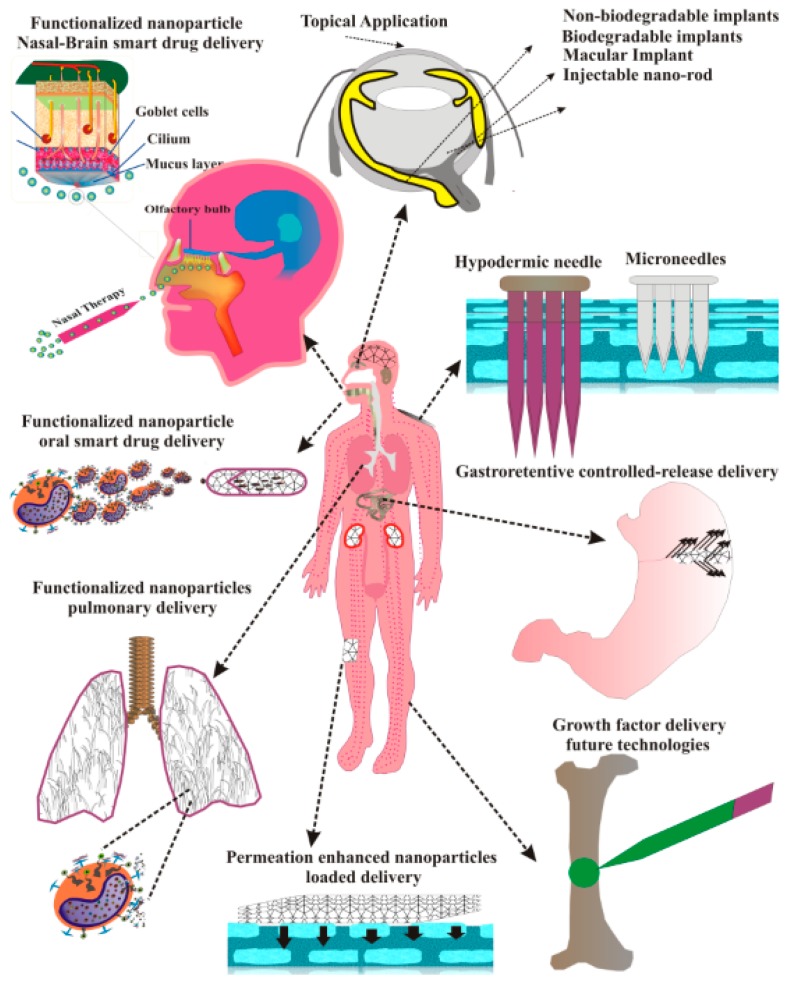
Schematic illustration of non-invasive or minimally invasive routes of administration and targeting strategies for polymeric nanoparticles. From Reference [191], open access peer-reviewed edited volume, Copyright (2014) IntechOpen.

**Figure 12 polymers-11-00745-f012:**
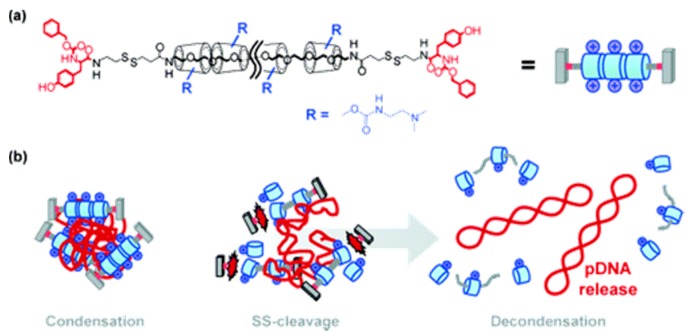
Chemical structure of biocleavable polyrotaxane: (**a**) polyplex formation and (**b**) terminal cleavage-triggered de-condensation of the polyplex. Reprinted with permission from Reference [210]. Copyright 2006 American Chemical Association.

**Table 1 polymers-11-00745-t001:** Major aspects of green nanotechnology and their practical implications for polymer nanoscience.

Green Nanotechnology Aspects	Practical Implication of the Aspect
1. Waste prevention	- Prevent intensive purification steps that utilizes harsh solvents and create chemical waste; Reference [32] enlists the solvents that meet the criteria for green chemistry - Use of more sophisticated purification techniques like nanofiltration to minimize solvent use - Bottom-up approach of PNP design is favored for fulfilling these aspects
2. Safer solvents and auxiliary reagents	
3. Reduce derivatives	
4. Atom economy	- Compact synthetic procedures with less steps involved - Use of catalysts to achieve selective reaction chemistries - Adjusting process parameters to reduce wastage of starting materials (atom efficiency) - Reduced production of by-products - Process monitoring at all steps involving complex PNP designs
5. Catalysis	
6. Process monitoring and control	
7. Design of biologically safe nanoplatforms	- Monitoring and adjustment of physical and chemical characteristics of PNPs, minimizing physiological toxicity - Chemical modification of NPs with polymers to reduce their innate toxicity
8. Use of natural/renewable raw materials	- Use of safe naturally occurring polymers for design of PNPs - Use of natural compounds like starch, proteins, sugars, and ascorbic acid, or safe synthetic compounds like PEG for NP coating to reduce toxicity - Use biodegradable polymers that degrade to harmless subunits easily removed by body’s defense mechanisms - Avoid accumulation of degradation remnants in the biological chain
9. Design for self-degradation	
10. Safer reaction chemistry	- Use of procedures and reagents with pre-defined safety - Identify replacement for highly toxic or pyrophoric reagents - Favor reaction chemistries that can be done at ambient temperatures - Ensure lowest possible reaction times to allow minimal exposure to chemical conditions - Utilize purest reagents and solvent systems
11. Energy efficient process of synthesis and maintaining stability	

**Table 2 polymers-11-00745-t002:** Comparison of average particle size and polydispersity index (PDI) between various synthesis methods.

Method of Polymer Synthesis	Example	Average Particle Size (nm)	PDI	Ref.
Solvent evaporation method	PLA NPs	200	<0.1	[48]
Solvent displacement method	PLGA NPs	160–170	<0.2	[56]
Salting out	PTMC NPs	184 ± 3	0.21	[60]
Single emulsion	PTMC NPs	334 ± 4	0.17	[60]
Dialysis	PLA sphere NPs	198.6	0.062	[66]
Supercritical fluid technology (RESS)	Raloxifene NPs	18.93 ± 3.73	<0.1	[19]
Supercritical fluid technology (RESOLV)	PHDFDA NPs	<50	<0.25	[77]
Recombinant technology	K8-ELP(1-60) NPs	<115	<0.2	[82]

**Table 3 polymers-11-00745-t003:** Advantages and challenges for PEGylation of nanoparticles.

Advantages	Challenges
(1) Improved blood circulation time and efficiency (2) Reduced tendency of opsonization leading to reduced recognition by macrophages (3) Capability to modulate NP size (4) Increased hydrophilicity of the carrier (5) Reduced liver accumulation (6) Modulation of pharmacokinetic parameters (dose reduction) (7) Reduced cellular and organ level toxicity (8) Improved tumor targeting (9) Improved cellular uptake (10) Large molecular weight PEG polymers led to increased half-lived of coated NPs	(1) Significant behavioral dependence on particle size. (2) Impact on pharmacodynamic properties of coated NP/gene complexes due to stearic hindrance at the binding site (3) Lack of conjugation sites and poor options for copolymerization (4) Likelihood of aggregation (5) Immunogenicity of PEG molecules in selective patient population (6) Partial fragmentation of proteins giving rise to new epitopes (7) Difficult characterization (8) Byproducts of reaction chemistries can affect sensitive therapeutic moieties (9) Increased risk of toxicity secondary to: (i) Renal or hepatic insufficiency (ii) Interactions within multiple-drug regimen

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
