# Peer review of "Polymeric Nanoparticles in Gene Therapy: New Avenues of Design and Optimization for Delivery Applications"

_polymers, 2019, doi:10.3390/polym11040745_

Round 1
Reviewer 1 Report
This is an interesting work. This manuscript summarized the applications of polymers in gene therapy, including the properties, designing strategies and challenges of polymers administrated in gene therapy. The distributions and cellular interaction of polyplexes and some non-invasive routes for polyplexes delivery are also included in this manuscript.
It remains some concerns.
1. It is a bit strange that there are no figures or tables in this review, as usually a comprehensive review is expected to show a summary figure or figures to give representative examples for each sections. I will recommend the addition of some pictures and tables, to summarize the structures or properties of different polymers for gene delivery applications.
2. In part 3.1 and 3.2, I am a bit curious to check the reason of including the synthesis strategies of nanoparticles, as they might not directly relate with gene delivery.
3. Besides to traditional polymeric non-viral gene carriers, recently polymeric hydrogels have also been reported to act as carriers with gene sustained release ability (i.e. Materials Chemistry Frontiers, 2019, DOI: 10.1039/C8QM00570B; Macromolecular Rapid Communications, 2018, 39, DOI: 10.1002/marc.201800117; Advanced Healthcare Materials, 2017, DOI: 10.1002/adhm.201700159; Materials Science and Engineering: C, 2018, DOI: 10.1016/j.msec.2017.08.075), which might be included in this review.
4. In chapter 7, it is worthy to put some prospective of polymers in gene therapy after conclusion part, which could be more valuable for readers.
5. Please proof-read the manuscript again as there are some spelling errors.
Author Response
We thank the reviewer for the constructive feedback
This is an interesting work. This manuscript summarized the applications of polymers in gene therapy, including the properties, designing strategies and challenges of polymers administrated in gene therapy. The distributions and cellular interaction of polyplexes and some non-invasive routes for polyplexes delivery are also included in this manuscript.
It remains some concerns.
1. It is a bit strange that there are no figures or tables in this review, as usually a comprehensive review is expected to show a summary figure or figures to give representative examples for each sections. I will recommend the addition of some pictures and tables, to summarize the structures or properties of different polymers for gene delivery applications.
Response: Figures and tables have been added.
2. In part 3.1 and 3.2, I am a bit curious to check the reason of including the synthesis strategies of nanoparticles, as they might not directly relate with gene delivery.
Response: Synthetic approaches crucially define the entrapment efficiency and surface chemistry of nanoparticles. Moreover, in most cases polymeric nanoparticles are designed in house for applications including gene therapy. For example, most top-down strategies like solvent displacement method are simpler techniques for designing PNPs, but not very suitable for water-soluble moieties including genetic materials. On the other hand, recombinant technology for producing PNPs discussed in the paper provides an insight of improved methods to overcome major gene delivery challenges. Therefore, the review summarizes briefly the traditional and novel approaches to design common polymeric gene carriers. We intend to present this review as one-stop summary for readers working in this field. Text is added to explain the relevance and provide clarity and to build a reference. Lines 237-243 added
3. Besides to traditional polymeric non-viral gene carriers, recently polymeric hydrogels have also been reported to act as carriers with gene sustained release ability (i.e. Materials Chemistry Frontiers, 2019, DOI: 10.1039/C8QM00570B; Macromolecular Rapid Communications, 2018, 39, DOI: 10.1002/marc.201800117; Advanced Healthcare Materials, 2017, DOI: 10.1002/adhm.201700159; Materials Science and Engineering: C, 2018, DOI: 10.1016/j.msec.2017.08.075), which might be included in this review.
Response: The authors appreciate this recommendation. This section is briefly included in the review as section 7
4. In chapter 7, it is worthy to put some prospective of polymers in gene therapy after conclusion part, which could be more valuable for readers.
Response: Added lines 986-995
5. Please proof-read the manuscript again as there are some spelling errors.
Response: We checked the manuscript for spelling errors.
Reviewer 2 Report
Reviewer Comments
Co-first authors Rai and Alwani have produced a compelling review paper on PNPs. It covers a wide breadth of topics. After a few minor revisions, the paper will be ready for publication. The suggested revisions are designed to add details for the more knowledgeable Polymer readership and broaden the authors’ analyses.
1. Line 107: specify that the ester groups degradation via hydrolysis in the presence of water.
2. Line 115: specify which group in the PLA polymer hydrolyzes.
3. Line 118: specify which group in the PCA polymer degrades and the conditions needed.
4. Line 135 and 155: Give one example of a green vs conventional chemistry reaction, and what makes it more environmentally safe (e.g. solvent X is no longer needed in you do reaction Y)
5. Section 3: several categories of top-down and bottom-up nanoparticle manufacturing methods are described, but it would be useful to a reader if a comment can be made on the polydispersity of the resulting nanoparticles (e.g. polydispersity index). This is an important Chemistry, Manufacturing and Control (CMC) consideration for a first-in-human safety trial. This extra layer of detail will enhance the analysis presented in this review.
6. Section 3.1.5: not much detail is given to how nanoparticles are actually formed using supercritical fluid technology. A few sentences or an example of how the nanoparticle formation happens would be a welcome enhancement.
7. Section 3.3.5: It could be useful to mention an existing commercial gene transfection product based on PEI, such as jetPEI®
8. Section 3.3.6: useful to also mention that PLGA-containing products are FDA approved, which is another reason many people used it (convenience)
9. Section 4.2: Worth mentioning that the PNP payload can also influence (increase) toxicity. Also, the time and cost associated with the method development needed to track the PNPs and degradation products in an Adsorption, Distribution, Metabolism and Excretion (ADME) study is quite high. While unfortunate, this is a reality in the field and should be acknowledged as another barrier.
10. General comment 1: useful to mention that Alnylam’s Patisiran, an siRNA nanoparticle, was recently approved by the FDA and that’s giving more confidence to the entire field.
Author Response
We thank the reviewer for the constructive comments.
Co-first authors Rai and Alwani have produced a compelling review paper on PNPs. It covers a wide breadth of topics. After a few minor revisions, the paper will be ready for publication. The suggested revisions are designed to add details for the more knowledgeable Polymer readership and broaden the authors’ analyses.
1. Line 107: specify that the ester groups degradation via hydrolysis in the presence of water.
Response: Added lines 126-127
2. Line 115: specify which group in the PLA polymer hydrolyzes.
Response: Added lines 135-136
3. Line 118: specify which group in the PCA polymer degrades and the conditions needed.
Response: Added lines 142-143
4. Line 135 and 155: Give one example of a green vs conventional chemistry reaction, and what makes it more environmentally safe (e.g. solvent X is no longer needed in you do reaction Y)
Response: Table 1 is added to explain each aspect and its practical implication in polymer science.
5. Section 3: several categories of top-down and bottom-up nanoparticle manufacturing methods are described, but it would be useful to a reader if a comment can be made on the polydispersity of the resulting nanoparticles (e.g. polydispersity index). This is an important Chemistry, Manufacturing and Control (CMC) consideration for a first-in-human safety trial. This extra layer of detail will enhance the analysis presented in this review.
Response: Table 2 is added.
6. Section 3.1.5: not much detail is given to how nanoparticles are actually formed using supercritical fluid technology. A few sentences or an example of how the nanoparticle formation happens would be a welcome enhancement.
Response: Explanation is provided in lines 334-338 and 2 diagrams (Figures 6 and 7) added.
7. Section 3.3.5: It could be useful to mention an existing commercial gene transfection product based on PEI, such as jetPEI®
Response: The authors acknowledge the importance of this comment adding lines 521-525.
8. Section 3.3.6: useful to also mention that PLGA-containing products are FDA approved, which is another reason many people used it (convenience)
Response: The authors acknowledge the importance of this comment. Marketed formulations are added in this section and explanation is provided to build up the reference. Lines 539-554 added.
9. Section 4.2: Worth mentioning that the PNP payload can also influence (increase) toxicity.
Response: Explanation is provided at the end of section 4.2 (lines 651-655).
Also, the time and cost associated with the method development needed to track the PNPs and degradation products in an Adsorption, Distribution, Metabolism and Excretion (ADME) study is quite high. While unfortunate, this is a reality in the field and should be acknowledged as another barrier.
Paragraph is added after section 4.3 to shed some light on this challenge in lines 689-702.
10. General comment 1: useful to mention that Alnylam’s Patisiran, an siRNA nanoparticle, was recently approved by the FDA and that’s giving more confidence to the entire field.
Response: Lines 59-63 are added to mention this technology.
Reviewer 3 Report
This paper is a quite comprehensive review of using polymers for non-viral gene delivery systems with a focus on synthetic polymer nanoparticles. The subject of the review is quite interesting; it is well-written in an engaging and lively style. In my opinion, this review can be considered for publication in Polymers after addressing the following comments:
1. The Title should reflect that the review focuses on polymeric nanoparticles for gene delivery.
2. In the Introduction, it is worthwhile to mention how this review differs from other recent reviews on this topic. This will allow authors to describe the aims of the review more clearly.
3. Section 2. ‘Properties of polymeric nanoparticles advantageous for biomedical use’ should be expanded with other properties (other than biodegradability and facile synthesis) that allow certain polymers to be used in gene therapy.
4. The authors describe in detail polymerization chemistries for common synthetic polymers (Section 3.3.). However, the discussion of synthetic routes for the preparation of semi-synthetic polymers of natural origin (such as chitin/chitosan, dextran, and their derivatives) is missed. I would recommend the authors to add this information to the review that would increase its value.
Author Response
We thank the reviewer for the constructive comments.
This paper is a quite comprehensive review of using polymers for non-viral gene delivery systems with a focus on synthetic polymer nanoparticles. The subject of the review is quite interesting; it is well-written in an engaging and lively style. In my opinion, this review can be considered for publication in Polymers after addressing the following comments:
1. The Title should reflect that the review focuses on polymeric nanoparticles for gene delivery.
Response: Title is revised.
2. In the Introduction, it is worthwhile to mention how this review differs from other recent reviews on this topic. This will allow authors to describe the aims of the review more clearly.
Response: Lines 83-87 address this comment.
3. Section 2. ‘Properties of polymeric nanoparticles advantageous for biomedical use’ should be expanded with other properties (other than biodegradability and facile synthesis) that allow certain polymers to be used in gene therapy.
Response: The authors acknowledge this comment. The whole section is revised. Different aspects of facile chemistry are covered. Table to define parameters of green nanomedicine with respect to polymer sciences is added. Lastly, section 2.3 scalable production is added
4. The authors describe in detail polymerization chemistries for common synthetic polymers (Section 3.3.). However, the discussion of synthetic routes for the preparation of semi-synthetic polymers of natural origin (such as chitin/chitosan, dextran, and their derivatives) is missed. I would recommend the authors to add this information to the review that would increase its value.
Response: This topic have been briefly highlighted as a part of revision (lines 248-261 added). For the sake of keeping the review comprehensive details are avoided.
Round 2
Reviewer 1 Report
I will recommend the publication of this revised version, and it will be interesting to many readers in the field of gene therapy.
Author Response
Thank you your time.
Reviewer 3 Report
The authors have successfully addressed the reviewers’ concerns, improving the manuscript with their edits. I understand the desire of the authors not to expand the review by details of using the natural or semi-synthetic polymers for gene delivery systems (lines 248-261). Therefore, I advise the authors to consider citing the most recent reviews on this topic (e.g. Kritchenkov et al RUSSIAN CHEMICAL REVIEWS 2017, 86:231–239, DOI: 10.1070/RCR4636; Santos-Carballal et al POLYMERS 2018, 10:444, DOI: 10.3390/polym10040444). It would help the readers to better understand the topic.
Author Response
Thank you for the suggestions; references were added.